# Genetic diversity and population structure of soybean (*Glycine max* (L.) Merril) germplasm

Tenena Silue[1,2]*, Paterne Angelot Agre[2], Bunmi Olasanmi[3], Adeyinka Saburi Adewumi[2], Idris Ishola Adejumobi[2], Abush Tesfaye Abebe[2]*

**1** Pan African University Life and Earth Sciences Institute (including Health and Agriculture), University of Ibadan, Ibadan, Oyo State, Nigeria, **2** International Institute of Tropical Agriculture (IITA), Ibadan, Oyo State, Nigeria, **3** Department of Crop and Horticultural Sciences, University of Ibadan, Ibadan, Oyo State, Nigeria

* at.abebe@cgiar.org (ATA); siluetenenan@yahoo.fr (TS)

## Abstract

Soybean (*Glycine max* (L.) Merril) is a significant legume crop for oil and protein. However, its yield in Africa is less than half the global average resulting in low production, which is inadequate for satisfying the continent's needs. To address this disparity in productivity, it is crucial to develop new high-yielding cultivars by utilizing the genetic diversity of existing germplasms. Consequently, the genetic diversity and population structure of various soybean accessions were evaluated in this study. To achieve this objective, a collection of 147 soybean accessions was genotyped using the Diversity Array Technology Sequencing method, enabling high-throughput analysis of 7,083 high-quality single-nucleotide polymorphisms (SNPs) distributed across the soybean genome. The average values observed for polymorphism information content (PIC), minor allele frequency, expected heterozygosity and observed heterozygosity were 0.277, 0.254, 0.344, and 0.110, respectively. The soybean genotypes were categorized into four groups on the basis of model-based population structure, principal component analysis, and discriminant analysis of the principal component. Alternatively, hierarchical clustering was used to organize the accessions into three distinct clusters. Analysis of molecular variance indicated that the genetic variance (77%) within the populations exceeded the variance (23%) among them. The insights gained from this study will assist breeders in selecting parental lines for genetic recombination. The present study demonstrates that soybean improvement is viable within the IITA breeding program, and its outcome will help to optimize the genetic enhancement of soybeans.

## Introduction

Soybean (*Glycine max* (L.) Merril) is a self-pollinated crop from the Fabaceae family with a diploid chromosome number of 2n = 40 [1]. It is one of the world's major

**Data availability statement:** All relevant data are within the manuscript and its Supporting Information files.

**Funding:** The field experiment was funded by 'IITA/USAID Genetic Improvement in Soy project, grant number PJ-2315 and Bill and Melinda Gates Foundation, grant number: Inv-046815-2023. These funds were received by Abush Tesfaye Abebe. The funders of this manuscript (IITA/USAID Genetic Improvement in Soy project, PJ-2315) and Bill and Melinda Gates Foundation had no role in the study design, data collection and analysis, decision to publish, or preparation of the manuscript. However, for manuscript payment, please refer to the following Bill and Melinda Gates Foundation grant: Inv-046815-2023.

**Competing interests:** The authors have declared that no competing interests exist.

legumes and oil crops in terms of production and trade [2]. Soybean contains approximately 38–42% high-quality protein and 18–20% oil rich in essential fatty acids [3]. In Nigeria, soybean is used to produce nutritious drinks known as "soymilk" and "awara" (soybean cake). It is also a crucial ingredient in poultry and fish feed and is also used in infant meals [4]. The oil is utilized in cooking and as a base for mayonnaise, margarine, salad dressings, and shortening [5].

Globally, soybean cultivation covers approximately 121 million hectares, with an estimated total production of 334 million tons annually [6]. The top three producers (Brazil, the United States, and Argentina) together contribute 73% of the world's production. In Africa, approximately 2.55 million hectares are dedicated to soybean cultivation, with an average productivity of 1,348 kg per hectare. South Africa, Nigeria, and Zambia are the leading producers on the continent, with annual production rates of 1.32, 0.73, and 0.35 million tons, respectively [7].

Soybean cultivation in Africa typically yields less than 1.5 t. ha$^{-1}$, which is significantly below the potential yield of over 3 t. ha$^{-1}$ [8]. This low productivity might be attributed to various factors, including the limited availability of high-yielding and climatically resilient improved varieties, poor soil fertility, diseases and pests, high pod shattering, inadequate agronomic practices, and particularly drought caused by inconsistent rainfall [9]. Therefore, there is a need for improved soybean varieties that are resilient to these biotic and abiotic stresses [10]. Furthermore, genetic diversity in many crops has decreased over time as commercial plant breeding focuses on enhancing one or a few traits and/or uses a limited number of exceptional genotypes to create a breeding population [11].

Exploiting and conserving crop genetic diversity is essential for developing new cultivars with desirable traits [12]. Assessing genetic diversity within germplasm is essential for expanding a core collection and enhancing germplasm utilization in breeding programs [2]. Additionally, understanding genetic variability within and between plant populations aids breeders in improving breeding strategies [13].

Crop variability can be assessed at both the phenotypic and genotypic levels using statistical methods to separate genetic and environmental components [14]. While morphological markers detect diversity, they are less effective than DNA markers because of their subjectivity, limited number, and environmental sensitivity [9]. DNA markers are more effective for evaluating genetic diversity, aiding in the efficient use of germplasms for conservation and crop yield improvement. Soybean genetic diversity has been assessed using various biochemical and molecular markers [15], including isozymes [16,17], random amplified polymorphic DNA (RAPD) [18], random fragment length polymorphism (RFLP) [19], amplified fragment length polymorphism (AFLP) [20], simple sequence repeats (SSR) [21–25], and single nucleotide polymorphisms (SNPs). Among these, SSR markers are effective for identifying genetic relationships within soybean populations [26–28], polymerase chain reaction (PCR) based amplification can sometimes result in sequence artefacts, complicating genotyping [29–34]. RAPD markers, although useful, have limitations like low discriminatory power and high genotyping costs [35]. The rise of

next-generation sequencing (NGS) has made SNP markers the preferred choice for studying genetic diversity due to their precision, cost-effectiveness, and even distribution across the genome [36,37].

According to Fischer et al. [38], SNP markers are the most effective among the molecular markers used as genomic resources for identifying variations in crop varieties, including soybeans. The high abundance in the genome and the ability to identify variation at a single locus make SNP markers outstanding among the marker groups explored for genomic activity [38]. Studies have generated extensive SNP data to explore genetic diversity between wild and cultivated soybeans, focusing on wild soybean allele diversity [39], uncovering valuable genetic information for breeding efforts [40], examining the genetic diversity and structure [41,42], and creating detailed haplotype maps using whole genome sequencing [43,44].

Diversity Array Technology (DArT) is a high-throughput genotyping method that provides cost-effective, scalable, whole-genome profiling, making it a versatile tool for various genetic applications [45]. It offers better coverage and fewer missing data compared to other next-generation sequencing (NGS) platforms and has been successfully used in crops like soybean [10,46,47], maize [48], wheat [49,50], cowpea and [51,52], sorghum [53], and garlic [54].

SNP markers, which are abundant and stable across the genome, are ideal for studying genetic variation and population structure [55]. While SNPs have been widely used to assess genetic diversity in soybeans globally, there is limited research on African germplasm, especially IITA's breeding materials. Understanding the genetic diversity of IITA's soybean germplasm can reveal distinct sub-populations, and historical breeding patterns, and guide future breeding efforts [56,57]. Population structure analysis is crucial for avoiding inbreeding, optimizing parent selection, and enhancing breeding outcomes. This study uniquely explores the genetic diversity of IITA's soybean germplasm using SNP markers, filling a critical gap in research on African soybean breeding populations.

Given the importance of genetic diversity assessment in optimizing breeding strategy for the IITA soybean improvement and the utility of SNP markers for improved precision for genetic diversity assessment, this study aims to assess the genetic diversity and population structure of IITA soybean germplasm using SNP markers. This will provide valuable insights for enhancing soybean breeding programs in SSA and contribute to the broader goal of improving food security in the region.

## Materials and methods

### Plant materials, planting, and leaf sampling and DNA extraction procedure

A total of 147 soybean accessions, comprised of 130 genotypes from the IITA soybean breeding program, 14 genotypes sourced from the United States Department of Agriculture (USDA) genetic resource center, and one genotype from Ghana, Uganda, and a private seed company (SeedCo) (list of germplasms, S1 Table), were selected and utilized for a molecular-based diversity assessment.

The 147 soybean accessions were sown and grown to the seedling stage in a screen house at IITA station Ibadan, Nigeria at 243 m.a.s., 7°30′8″N longitude and 3°54′37″E latitude. Three weeks after planting, five-leaf discs 5 mm in diameter from young and healthy leaves were collected via a biopsy curette from the leaf blades of each of the 147 genotypes. The leaf samples were placed into 96-well collection plates (12 x 8-strip tubes per 96-deep well plate) and lyophilized using a Labconco Freezone 6 plus dryer. The lyophilized leaf samples were sent to Diversity Array Technology (DArT)®, Canberra, Australia, for DNA extraction, library construction, and SNP marker development.

The DNA was extracted using a technique developed by Intertek-AgriTech (http://www.intertek.com/agriculture/agritech/) and based on the LGC oKtopure™ automated high-throughput 'sbeadex™' DNA extraction and purification system (https://www.biosearchtech.com/). Magnetic separation is used in the 'sbeadex™' technique to prepare nucleic acids. The first stage in this process is to homogenize leaf tissue samples in 96 deep-well plates using steel bead grinding. The ground tissue is treated with a DNA extraction buffer using LGC's 'sbeadex™' kit for plant DNA preparation

(https://www.biosearchtech.com/). Finally, super-paramagnetic particles coated with 'sbeadexTM' surface chemistry that catches nucleic acids from a sample are used to purify extracted DNA. Purified DNA is eluted and used in downstream procedures.

High-throughput genotyping was conducted in 96 plex DArTseq protocol, and SNPs were called using the DArT's proprietary software, DArTSoft, as described by Killian et al. [58]. Each sequencing result's reads and tags were aligned to the soybean reference genome [59].

### SNP marker quality control

Single-row format data from DArT were initially converted into HapMap and variant call format (VCF) formats using KDcompute (https://kdcompute.seqart.net/kdcompute, accessed on 07/06/2024). SNP-derived markers were then first filtered using PLINK 1.9 and VCFtools, on the basis of the call rate of the raw data [60]. The SNP markers with call rates ranging from 0.80 to 1.0 were selected for further quality control analyses. Duplicate SNP markers across the chromosomes were removed. This process involved removing markers with minor allele frequencies of less than 5%, markers and genotypes with more than 20% missing data, and those with a low coverage read depth of less than 5 [61,62].

### Statistical analyses

The structure and pattern of genetic diversity within soybean genotypes were assessed via genotypic data generated on the basis of SNP markers. VCFtools and PLINK 1.9 were used to estimate summary statistics such as observed and expected heterozygosity, minor allele frequency (MAF), and polymorphic information content (PIC). The genotypic data was formatted in dosage format (0,1,2) using the recodeA function in Plink, where 0 is the homozygote reference, 1 is the homozygote alternative, and 2 is the heterozygote. The generated dosage format was then analyzed with the vegan library in R to estimate several genetic diversity indices, including the Shannon-Wiener index (H′), the inverse Simpson index (1/D) and the Alpha diversity index (A). These indices were used to assess the soybean genotypes' overall genetic diversity and allelic richness, following the methodology outlined by Oksanen et al. [63]. The SNP distribution and density plot of the SNP markers across the 20 chromosomes of the soybean genome was constructed via the CMplot package [64]. The SNP markers data were subjected to population structure analysis following the method described by Agre et al. [65]. By testing cluster numbers ranging from 2 to 10, the optimal number of clusters was identified through k-means analysis, employing cross-validation on the basis of the Bayesian information criterion (BIC). Each soybean genotype was assigned to its respective cluster if it had at least 70% ancestry probability. Genotypes with less than 70% ancestry were considered as admixed. The diversity pattern revealed through population structure analysis was further supported by discriminant analysis of the principal component (DAPC) via the Adegenet package in R [66]. DAPC, which uses the k-means clustering method, aims to minimize variance within clusters while maximizing variance between clusters [67]. Pairwise genetic dissimilarity distances (identity-by-state, IBS) were calculated via the Jaccard method, implemented in the Philentropy R package [68]. A Ward's minimum variance hierarchical cluster dendrogram was then constructed from the Jaccard dissimilarity matrix using the Analyses of Phylogenetics and Evolution (APE) package in R [69]. Principal component analysis (PCA) was subsequently conducted to determine the genetic relationships among 147 soybean genotypes via the FactoMineR [70] and FactoExtra R packages [71]. Molecular variance analysis (AMOVA) and calculation of the coefficient of genetic differentiation among populations (PhiPT) were performed to investigate the distribution of genetic diversity among and within hierarchical populations via GenAlEx software (v.6.51) [72].

## Results

### Genetic diversity indices

A total of 59,126 SNP markers from the 147 soybean genotypes were originally generated via the Diversity Arrays Technology (DArT) platform. The transformation of these allelic sequences into genotypic data resulted in a raw data file of

53,418 SNPs, and after quality control analysis (SNP filtering), only 7,083 SNP markers were retained for further analyses. These markers were unequally distributed across the 20 soybean chromosomes (Fig 1, Table 1). The genome-wide SNP density plot indicated that chromosome 18 had the highest concentration of SNPs, accounting for 7.6% of the total number of markers with 538 SNPs. In contrast, chromosome 12 had the lowest concentration, with only 3.17% of the SNPs, totaling 225 markers (Fig 1). The diversity indices for the SNP marker presented a polymorphic information content (PIC) value of 0.277, ranging from 0.262 to 0.293. The MAF averaged 0.254 across all the markers. The observed heterozygosity (Ho) ranged from 0.093 to 0.124, with an average value of 0.110. The expected heterozygosity (He) varied between 0.322 and 0.371, with an average of 0.344 (Table 1 and S1 Fig). The Shannon diversity index (H′) index ranged from 8.505 to 8.704, with a mean value of 8.597. The inverse Simpon's index (1/D) had an average of 5349.145, with a range from 4903.021 to 5824.754. The Alpha diversity index (A) varied between 4218.816 and 7568.063, with a mean value of 4950.72.

## Population structure of 147 soybean breeding lines

Various complementary methods, including a model-based Bayesian approach in ADMIXTURE, DAPC, and PCA), were utilized to analyze the population structure of the 147 soybean accessions. Based on the admixture results, four subpopulations (K=4) were identified (Fig 2). Similarly, DAPC revealed four genetic groups (Fig 3), following a sharp decline in the Bayesian information criterion (BIC) versus the number of cluster plots (S2 Fig). There was a disparity in how the soybean accessions were assigned to the identified genetic groups between the ADMIXTURE and DAPC results. This disparity may be related to the DAPC analysis, which assigned the 147 soybean genotypes into distinct groups. In contrast, ADMIXTURE assigned 57% of the accessions (84 genotypes) to the four subpopulations on the basis of a membership probability of 70%, whereas the remaining 43% (63 lines) of the collection were classed as admixtures.

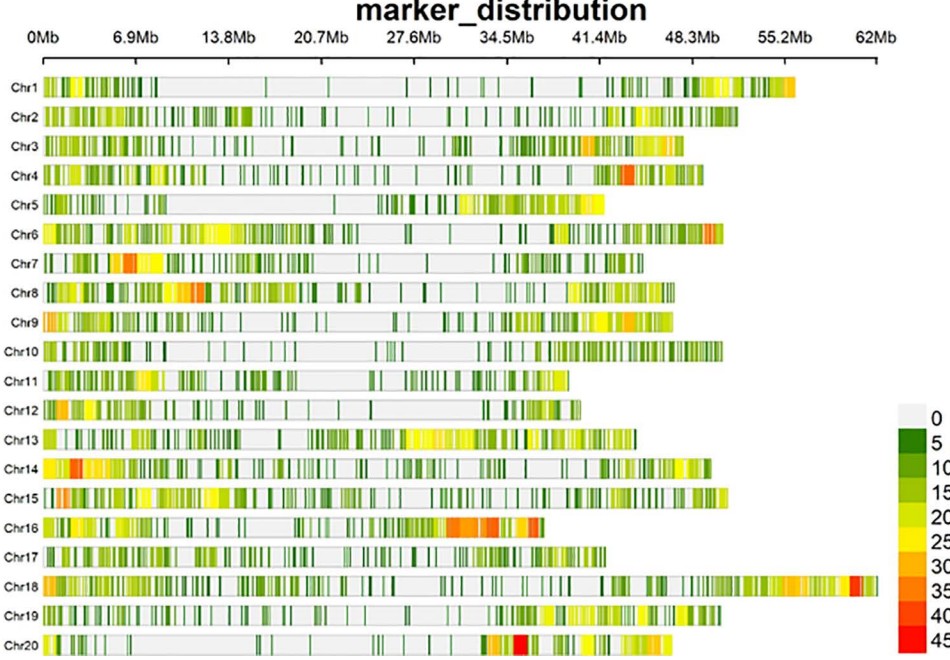

**Fig 1. Distribution and density of filtered SNPs across 20 soybean chromosomes.** The horizontal axis represents the chromosome length, the SNP density in each region is indicated at the bottom right.

**Table 1. Summary statistics of diversity indices of 147 soybean accessions based on SNP markers.**

| Chr | Nbr of SNPs | Ho | He | MAF | PIC | H′ | 1/D | A |
|---|---|---|---|---|---|---|---|---|
| 1 | 260 | 0.116 | 0.359 | 0.267 | 0.287 | 5.304 | 198.66 | 188.77 |
| 2 | 334 | 0.111 | 0.335 | 0.250 | 0.269 | 5.546 | 253.235 | 235.059 |
| 3 | 345 | 0.105 | 0.338 | 0.246 | 0.272 | 5.579 | 262.490 | 242.013 |
| 4 | 328 | 0.094 | 0.322 | 0.231 | 0.262 | 5.555 | 256.267 | 231.879 |
| 5 | 285 | 0.104 | 0.349 | 0.256 | 0.281 | 5.382 | 215.95 | 198.54 |
| 6 | 462 | 0.095 | 0.34 | 0.251 | 0.274 | 5.886 | 356.442 | 325.612 |
| 7 | 296 | 0.093 | 0.323 | 0.230 | 0.263 | 5.438 | 228.309 | 205.593 |
| 8 | 440 | 0.108 | 0.334 | 0.241 | 0.270 | 5.838 | 339.115 | 313.574 |
| 9 | 374 | 0.108 | 0.371 | 0.288 | 0.293 | 5.619 | 271.903 | 256.762 |
| 10 | 247 | 0.108 | 0.348 | 0.259 | 0.278 | 5.225 | 184.079 | 170.973 |
| 11 | 274 | 0.105 | 0.339 | 0.250 | 0.272 | 5.340 | 206.522 | 189.444 |
| 12 | 225 | 0.113 | 0.354 | 0.265 | 0.284 | 5.139 | 168.733 | 157.169 |
| 13 | 470 | 0.124 | 0.357 | 0.273 | 0.284 | 5.870 | 349.287 | 332.682 |
| 14 | 399 | 0.112 | 0.357 | 0.265 | 0.287 | 5.712 | 299.103 | 332.682 |
| 15 | 435 | 0.112 | 0.345 | 0.249 | 0.278 | 5.820 | 333.216 | 310.035 |
| 16 | 463 | 0.119 | 0.333 | 0.238 | 0.270 | 5.876 | 352.129 | 326.264 |
| 17 | 290 | 0.124 | 0.357 | 0.269 | 0.284 | 5.405 | 220.020 | 207.293 |
| 18 | 538 | 0.114 | 0.344 | 0.255 | 0.277 | 6.002 | 399.211 | 37098 |
| 19 | 310 | 0.112 | 0.353 | 0.232 | 0.284 | 5.418 | 223.069 | 208.533 |
| 20 | 308 | 0.111 | 0.336 | 0.250 | 0.271 | 5.460 | 232.811 | 216.033 |
| **Min** | 225 | 0.094 | 0.322 | 0.230 | 0.262 | 8.505 | 4903.021 | 4218.816 |
| **Mean** | 354.15 | 0.110 | 0.344 | 0.254 | 0.277 | 8.597 | 5349.145 | 4950.728 |
| **Max** | 538 | 0.124 | 0.371 | 0.228 | 0.293 | 8.704 | 5824.754 | 7568.063 |

Chr: chromosome, Nbr of SNPs: Number of single nucleotide polymorphism, Ho: observed heterozygosity; He: expected heterozygosity; MAF: minor allelic frequency; PIC: polymorphism information content, H′: Shannon-Wiener index, 1/D: inverse Simpson index, A: Alpha diversity index, Max= maximum, Min= minimum

Hierarchical cluster (HC) analysis grouped all 147 soybean genotypes into three major genetic groups or clusters (Fig 4 and S3 Fig). Cluster 1 contained 87 genotypes, predominantly IITA breeding lines, except a single genotype, 'SONGDA' introduced from Ghana, which was originally an IITA breeding line sent to Ghana in variety trials. The 86 IITA genotypes were mainly TGx (Tropical *Glycine max*) varieties or progenies resulting from crosses between two TGx parental lines (S1 Table). The HC analysis grouped these 87 genotypes into Cluster 1, while the DAPC divided them into two distinct clusters, represented as Clusters 3 and 4 (Fig 3). According to the ADMIXTURE analysis, 36 of the 87 genotypes in Cluster 1, including the unique Ghana genotype, were classified as admixtures. The remaining 51 accessions were assigned to the blue and cyan groups, with 34 and 17 genotypes, respectively (Fig 2). Cluster 2 comprised 34 accessions, including 16 IITA-breeding lines, 16 of the 17 genotypes sourced from the USDA soybean genetic resource center, and one variety Sc-Signa from SeedCo (a private Company) and MAKSOY-4N from Makere University, Uganda. The 16 IITA breeding lines consisted of progenies derived from various parental lines, including TGx, ZIGx, SPSOY, CIMARRONA, PI567090, SOYICA and ST SUPREMA (S1 Table). Among the 34 genotypes in Cluster 2 identified by HC analysis, 15, exclusively IITA breeding lines, were clustered by ADMIXTURE analysis in subpopulation 1 (red) (Fig 2). The remaining 19 genotypes, which included one IITA breeding line, 16 from Columbia, and the unique genotypes from SeedCo and Uganda, were assigned as admixes by ADMIXTURE analysis. On the other hand, the DAPC analysis placed all Cluster 2 genotypes into

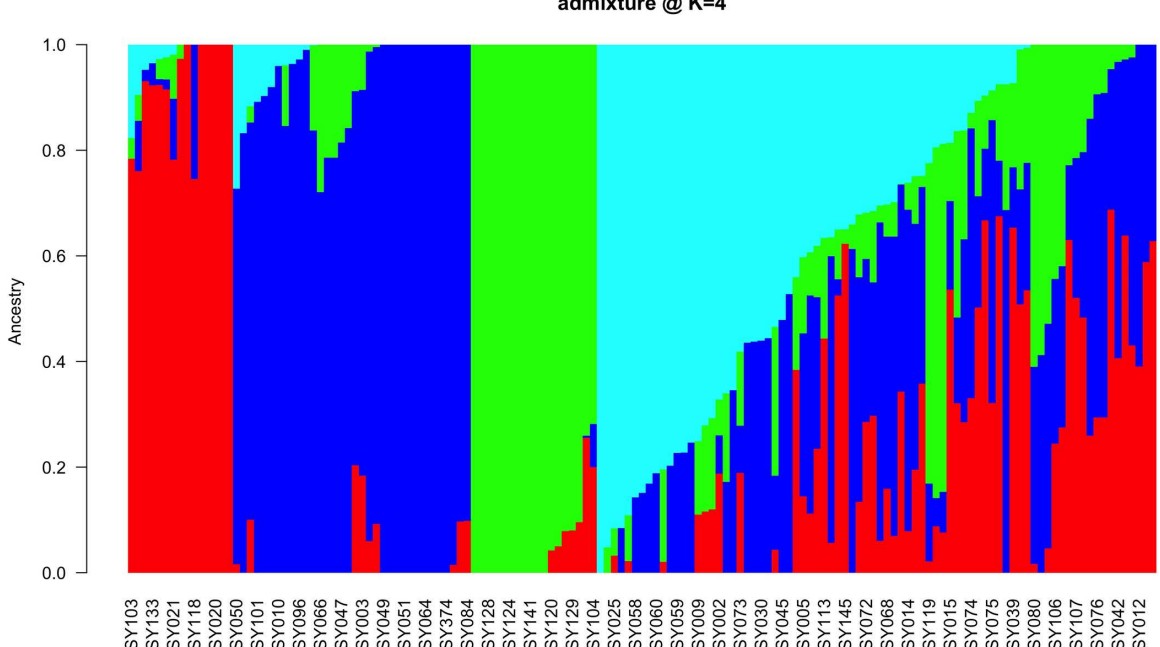

**Fig 2. Population structure of 147 soybean breeding lines from the IITA breeding program, Ibadan on the basis of ADMIXTURE analysis with the subpopulations set at K = 4 via 7,083 high-quality SNPs.** The colors correspond to the four subpopulations: Subpopulation 1 (red), Subpopulation 2 (blue), Subpopulation 3 (green) and Subpopulation 4 (cyan), determined by a membership coefficient greater than70%.

Cluster 1 (Fig 3). The 26 genotypes assigned to Cluster 3 by HC analysis included 25 IITA breeding lines and one Columbia genotype (Panaroma-3). The IITA breeding lines were a mix of pure TGx parents and backcross progenies, derived from crosses between TGx lines and other parental lines, such as ST SUPREMA, CIMARONA, SOYICA, ZIGx, LG-12, and AS-G (S1 Table). The DAPC analysis classified these 26 genotypes from Cluster 3 in the HC analysis into Cluster 2 (Fig 3). Moreover, the ADMIXTURE analysis placed 18 of them into a specific group (green) (Fig 3), whereas the remaining 8, including the unique USDA genotype (Panaroma-3), were categorized as admixtures.

Principal component analysis (PCA) revealed that the first and second components (PC1 and PC2) accounted for 45.2% and 15.9% of the total molecular variation, respectively, together explaining 61.1% of the overall observed variation (Fig 5). Although all the genotypes within each cluster were grouped, they exhibited some heterogeneity. The genotypes classified as admixtures were identified as admixed groups in the PCA (Fig 5).

### Genetic distance and differentiation of soybean accessions

A pairwise dissimilarity genetic distance matrix revealed that the genetic distance among the 147 soybean genotypes ranged from 0.012 to 0.452, with an average distance of 0.333. The greatest genetic distance of 0.452 was found between the USDA genotype TGx 2029-39F (Cluster 2) and two IITA breeding lines, TGx 2002–89 GN and TGx1988-5FxTGx1989-19F-9, both in Cluster 1. In contrast, the lowest genetic distance (0.012) was observed between two IITA lines, TGx 2002–89 GN and TGx 2002–90 GN, both of which belonged to cluster 1. Within Cluster 1, the genetic distances ranged from 0.012 to 0.452 with an average of 0.337. Cluster 2 presented genetic distances ranging from 0.015 to 0.452, with an average of 0.367. For Cluster 3, the distances ranged from 0.017 to 0.433, with an average of 0.355.

The analysis of molecular variance (AMOVA) revealed that 77% of the total genetic variability was partitioned as within-population variation, which was significantly greater than the 23% partitioned among among-populations variation

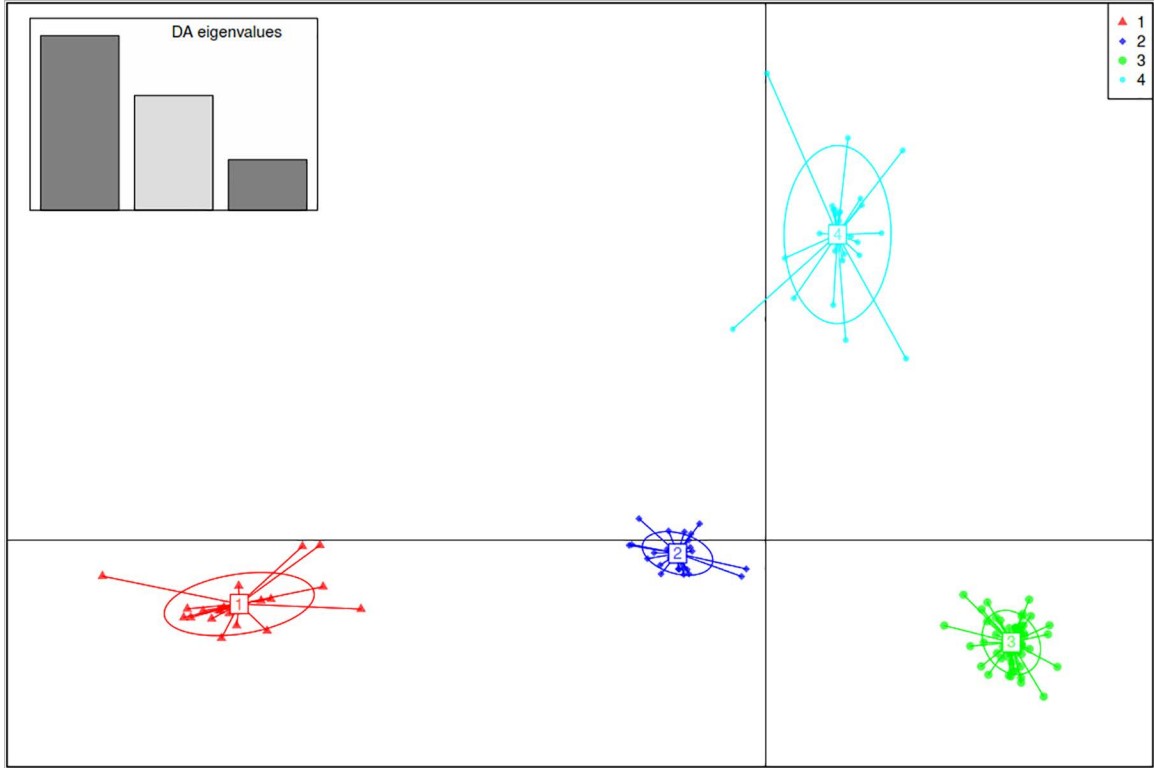

**Fig 3. Summary of discriminant analysis of principal component (DAPC) for 147 soybean accessions, illustrating the ordination plot of DAPC for the four groups.** Eigenvalues are displayed in the upper-left inset. Genetic groups or clusters are represented by distinct colors and inertia ellipses, with individual genotypes indicated by dots.

(Table 2). The overall genetic differentiation (PhiPT) and gene flow (Nm) for the 147 soybean genotypes were 0.233 (p < 0.001) and 1.649, respectively.

The pairwise population differentiation (PhiPT) estimates revealed that the highest degree of differentiation (0.267) was observed between populations 1 and 3, whereas the lowest degree of differentiation (0.200) occurred between populations 1 and 2. The genetic differentiation between population 2 and population 3 was 0.244. On the other hand, the pairwise population estimates of gene flow (Nm) for the three populations ranged from 1.376 to 1.998 migrants per population (Table 3).

## Discussion

Studying the genetic diversity of germplasm or breeding material is the best approach for understanding the existing genetic variation and effectively managing genetic resources to enhance breeding programs [73,74]. Hence, plant breeders need such genetic analysis to execute strategic target selection and integration while maintaining significant economic traits associated with distinct crops [75].

The average value of 0.277 indicates that the markers used in this study were both informative and polymorphic. Given the bi-allelic nature of SNPs, where the PIC cannot exceed 0.5 [76], the PIC values observed in this study are suitable for differentiating the 147 soybean accessions. Similar results have been reported in soybean studies, including Abebe et al. [2], who found a mean PIC value of 0.25 for elite soybean lines developed by IITA, and Lee et al. [77], who reported a PIC value of 0.22 when evaluating 228 soybean genotypes. Tsindi et al. [78] also reported a PIC value of 0.2 for 210 South

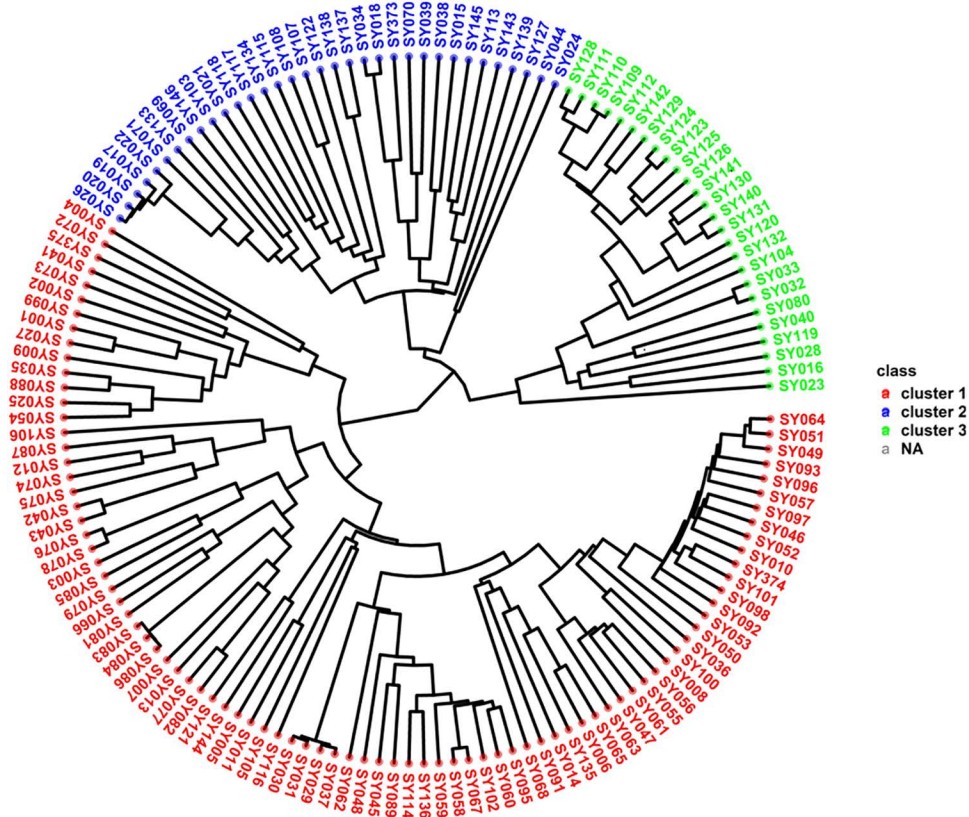

**Fig4. Hierarchical clustering analysis based on 7,083 DArT-SNP markers, depicting the genetic relationships among 147 soybean accessions from the IITA, Ibadan breeding programme.**

African soybean genotypes. In other self-pollinated crops, Singh et al. [76] reported a mean PIC value of 0.23 in rice. This study also demonstrated the possibility of using the selected DArTseq-SNP markers for genomic investigations in soybeans, which may serve as a foundation for future breeding efforts in the IITA soybean breeding program and conservation initiatives in Nigeria. The MAF value measures the selective ability of the marker. Owing to the bi-allelic nature of SNP markers, the MAF closest to 0.5, is best. The high average MAF value of 0.254 observed in this study indicates valuable genes can be exploited from those genotypes [75]. Compared to the results based on SNP markers reported by Hao et al. [79], our MAF values are lower. Their study revealed that the MAF ranged from 0.102 to 0.50 in soybean landraces, with an average value of 0.291. This difference might be because the materials used in the present study were advanced breeding lines, whereas Hao et al. [79] focused mainly on landraces. The average expected heterozygosity (He) of 0.344 indicates high genetic diversity within the soybean accessions, which can be effectively used for soybean improvement [75]. The Shannon-Wiever diversity index (H′), which quantifies the entropy or uncertainty in the genetic composition of a population, has a mean value of 8.597. This relatively high value indicates a genetically diverse population with various alleles and genotypes. Such a result suggests that our soybean genotypes are built on a genetic foundation, with a broad range of genetic types contributing to its overall diversity. Similarly, the inverse Simpson index (1/D), with its high mean value of 5349.145, reinforces the genetic diversity within our soybean population. It also implies a balanced distribution of genotypes without any single genotype dominating, reflecting a well-represented mix of genetic types. Comparable findings have been reported in studies on soybean [80,81], maize [82], rice [83], and wheat [84]. Furthermore, the alpha

**Fig 5. Principal component analysis plot showing the clustering of 147 soybean breeding accessions into four clusters.** Each cluster is represented by a distinct color: Cluster 1 (red), Cluster 2 (yellow), Cluster 3 (green), Cluster 4 (blue), and admixed individuals (pink).

**Table 2. Analysis of molecular variance (AMOVA), population differentiation (PhiPT), and gene flow (Nm) within and among soybean populations.**

| Source of Variation | | DF | SS | MS | Est. Var. | % |
|---|---|---|---|---|---|---|
| Within populations | | 144 | 184936.218 | 1284.279 | 1284.279 | 77 |
| Among populations | | 2 | 34906.864 | 17453.432 | 389.395 | 23 |
| Total | | 146 | 219843.082 | | 1673.674 | 100 |
| Statistical test | PhiPT | | | | 0.233 | |
| | *p* (rand>= data) | | | | 0.001 | |
| | Nm (haploid) | | | | 1.649 | |

DF, degrees of freedom; SS, sum of squares; MS, mean square deviation; Est.Var., estimated variance component; percentage of the total variance (%) contributed by each component and significance of variance (p-value); Nm, gene flow.

diversity index (A), which assesses both the richness (the number of distinct genotypes or alleles) and evenness (the relative abundance of each genotype/allele) within a population, also exhibits a high mean value (4950.720). This suggests that while genetic diversity may vary across different subpopulations or sites, the overall population remains genetically rich, as reported by Adejumobi et al [85].

Analyzing population structure via SNP markers offers helpful information for preserving and tracking the genetic diversity essential for an effective breeding program [86]. ADMIXTURE and DAPC analyses were used to determine the

**Table 3. Pairwise population comparison using population differentiation (PhiPT), and gene flow (Nm) values based on 999 permutations from AMOVA according to Hierarchical Cluster analysis.**

| Population 1 | Population 2 | PhiPT | Nm |
|---|---|---|---|
| Cluster 1 | Cluster 2 | 0.200*** | 1.998 |
| Cluster 1 | Cluster 3 | 0.267*** | 1.376 |
| Cluster 2 | Cluster 3 | 0.244*** | 1.546 |

***PhiPT values significantly greater than 0, p < 0.001.

population structure, revealing the presence of four major populations (K = 4) for the 147 soybean genotypes. However, previous studies [2] and [6] reported different ADMIXTURES results, with ΔK values of 3 and 6, respectively. Considerable levels of admixture (42.87%) were detected among the genotypes, which likely resulted from historical gene flow, breeding practices, and inherent genetic diversity within and between the soybean populations [56]. Chander et al. [7] reported similar levels of admixture in their study of 165 soybean genotypes, which primarily consisted of IITA-bred soybean varieties. In contrast to the results of the ADMIXTURE and DAPC analyses, the hierarchical cluster (HC) method classified the 147 genotypes into three major clusters, suggesting that this could represent the optimal number of genetic clusters within the soybean accessions studied [78]. These results highlight the effectiveness of SNP markers in identifying superior accessions that have the potential to enhance the genetic diversity of the soybean population [87]. Furthermore, the results emphasize the importance of the distinct pedigrees of the soybean genotypes in maintaining genetic variation, as genotypes with similar pedigrees tended to cluster together based on their SNP profiles. Similar clustering patterns, which reflect the genetic origin of the accessions, have been reported in soybean [79–88,89] as well as in other legume species such as cowpea [90,91] and sesame [92]. Grouping the 147 genotypes into four distinct clusters within the first two principal components accounting for more than 60% of the total genetic variation indicates a high level of variation among the genotypes across the clusters, but high relatedness within a specific cluster. These results suggest that the genotypes within a given cluster share significant genetic similarities, making them potentially valuable for enhancing the genetic diversity of soybean breeding programs through hybridization. In support of this, Bakayoko et al. [60] highlighted that genotypes within the same cluster are genetically similar and thus could play a crucial role in genetic improvement efforts.

The results from the analysis of molecular variance (AMOVA) suggest a significant level of gene flow, with 23% of the total variation attributable to differences among populations, while 77% of the variation was observed within populations. This indicates that the majority of genetic variation resides within populations, but there is still considerable variation between them, supporting the presence of gene flow. These results are further corroborated by the gene flow (Nm) value, which plays a key role in enhancing the genetic diversity and influencing genetic differentiation of plant populations and is a crucial factor influencing genetic differentiation [93]. When Nm is greater than 1, gene flow is sufficient to counteract the effects of genetic drift. In this study, the average gene flow value (Nm = 1.649) suggests that the soybean populations are not yet significantly impacted by genetic drift. Similar findings have been reported in previous studies on soybean [10,89,94,95] and other crops, such as Camelina sativa [96], wheat [97], rice [14], cowpea [74–98], and potatoes [99]. The PhiPT value (an analogue of the fixation index Fst) of 0.23 indicates a high level of genetic differentiation among populations, suggesting limited gene exchange. In general, low Fst values close to 0 suggest that subpopulations are genetically similar, with minimal divergence, whereas an Fst of 1 indicates complete genetic fixation within subpopulations [92–100]. This analysis also revealed significant genetic differentiation between populations 1 and 2 and between populations 2 and 3. Moreover, the differentiation between populations 1 and 3 was particularly pronounced. This substantial genetic diversity among all pairwise populations highlights the considerable diversity within the soybean accession and emphasizes the effectiveness of the selected markers for future research on soybean genetic diversity [2]. Consequently, hybridizing

genotypes from different populations could introduce valuable genetic variation, enhancing genetic gain through focused selection [10].

## Conclusion

The soybean lines analyzed in this study exhibit high levels of polymorphism and genetic diversity, reflecting considerable genetic variability within the population. A distinct genetic structure was observed among the sub-populations, which were grouped based on their pedigree or geographic origin. The distribution of soybean genotypes across major clusters or sub-populations, as revealed by multivariate analyses, highlights the success of IITA's plant breeding efforts in creating a diverse genetic base. This diversity has been achieved while maintaining a focus on enhancing local adaptation to various agroecological zones within soybean-growing areas of West Africa. The diverse nature of materials used in this study, suggests these materials serve as valuable sources of genetic variation. These genotypes potentially harbor contrasting parental traits and novel alleles relevant to economically significant characteristics such as yield, drought resistance, and pod shattering. These findings present an opportunity for soybean breeders to improve the efficient selection of parental lines. Moreover, the study emphasizes the potential need to integrate exotic germplasm into breeding programs to further enrich the genetic diversity base of soybeans in the region.

## Supporting information

**S1 Table. List of the soybean (*Glycine max* (L. ) Merril) accessions used in this study and their respective origins.** (DOCX)

**S1 Fig. Summary statistics of 7,083 single nucleotide polymorphism (SNP) markers used for genotyping 147 soybean accessions.** (a) Expected heterozygosity, (b) observed heterozygosity, (c) minor allele frequency and (d) polymorphic information content.
(TIF)

**S2 Fig. Graph showing the best k value via Bayesian information criterion analysis.**
(TIF)

**S3 Fig. Silhouette graph showing the optimal number of hierarchical clusters.**
(TIF)

## Acknowledgements

We gratefully acknowledge the technical support the soybean breeding team provided in establishing the trials and leaf sample collection.

## Author contributions

**Conceptualization:** Tenena Silue, Paterne Angelot Agre, Bunmi Olasanmi, Abush Tesfaye Abebe.

**Data curation:** Tenena Silue.

**Formal analysis:** Tenena Silue, Paterne Angelot Agre, Adeyinka Saburi Adewumi, Idris Ishola Adejumobi.

**Funding acquisition:** Abush Tesfaye Abebe.

**Investigation:** Tenena Silue.

**Methodology:** Tenena Silue, Paterne Angelot Agre, Adeyinka Saburi Adewumi, Abush Tesfaye Abebe.

**Project administration:** Abush Tesfaye Abebe.

**Resources:** Abush Tesfaye Abebe.

**Software:** Tenena Silue, Paterne Angelot Agre, Adeyinka Saburi Adewumi.

**Supervision:** Paterne Angelot Agre, Bunmi Olasanmi, Abush Tesfaye Abebe.

**Validation:** Tenena Silue, Paterne Angelot Agre, Adeyinka Saburi Adewumi, Abush Tesfaye Abebe.

**Visualization:** Tenena Silue, Paterne Angelot Agre, Adeyinka Saburi Adewumi.

**Writing – original draft:** Tenena Silue.

**Writing – review & editing:** Tenena Silue, Paterne Angelot Agre, Bunmi Olasanmi, Adeyinka Saburi Adewumi, Idris Ishola Adejumobi, Abush Tesfaye Abebe.

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
