## [Decision Letter · Decision Letter 0]

6 Nov 2024

PONE-D-24-43093Genetic diversity and population structure of soybean (Glycine max (L.) Merril) germplasm.PLOS ONE

Dear Dr. SILUE,

Thank you for submitting your manuscript to PLOS ONE. After careful consideration, we feel that it has merit but does not fully meet PLOS ONE’s publication criteria as it currently stands. Therefore, we invite you to submit a revised version of the manuscript that addresses the points raised during the review process.

We look forward to receiving your revised manuscript.

Kind regards,

Tzen-Yuh Chiang

Academic Editor

PLOS ONE

Journal Requirements:

2. Thank you for stating the following financial disclosure: [The field experiment was funded by 'IITA/USAID Genetic Improvement in Soy project, PJ-2315.]

Reviewers' comments:

Reviewer's Responses to Questions

**Comments to the Author**

1. Is the manuscript technically sound, and do the data support the conclusions?

Reviewer #1: Partly

2. Has the statistical analysis been performed appropriately and rigorously? 

Reviewer #1: No

3. Have the authors made all data underlying the findings in their manuscript fully available?

Reviewer #1: No

4. Is the manuscript presented in an intelligible fashion and written in standard English?

Reviewer #1: No

5. Review Comments to the Author

Reviewer #1: Dear authors

The researchers studied the genetic diversity of soybean by using DArT SNP technology. There isn't much new information on the subject of this study. The following adjustments are required:

Abstract

The value of AMOVA should be added

Keywords

The terms from the title should not be utilized as keywords. As a result, the keyword structure needs to be updated.

Introduction

Some information about the genetic diversity technique assessment should be provided.

Some information on the SNP markers should be provided.

More information about the DArt technology should be spotted

The researchers should supplement the hypothesis statement with a few words describing the knowledge gap that their research addresses.

In their conclusion, the authors should make a statement about their originality.

What additional research-related acts or relationships did the authors discover in this study compared to previous ones?

The overall and specific objectives should be adequately documented.

Materials and Methods

All abbreviations should be written in detail name.

The DNA extraction and PCR procedure should be detailed

References should support all procedures.

The DArt procedure should be written in depth.

All materials' manufacturing processes should be documented.

The criteria for conducting population structure analysis should be mentioned.

Results and discussion

The contents of all tables should be detailed.

The genetic diversity index for population should be calculated

All abbreviations should be defined in the captions of tables

The explanation is unconvincing. The discussion section primarily repeats findings rather than critically deconstructing them, resulting in a subpar presentation that deserves improvement. The writers should explain how all of the investigation's findings relate to their own conclusions. The researchers must look into and analyze the outcomes of PCA, AHC, PIC, Fst, AMOVA, and diversity index. They should provide a thorough explanation of the implications of high or low readings for each of these variables.

Conclusion

This section does not include a conclusion. Given the accessibility of this section, authors must provide a brief summary of the key findings. Future research on this area should delve deeper.

Best regards

6. PLOS authors have the option to publish the peer review history of their article (what does this mean? ). If published, this will include your full peer review and any attached files.

**Do you want your identity to be public for this peer review?** For information about this choice, including consent withdrawal, please see our Privacy Policy .

Reviewer #1: **Yes: ** Nawroz Tahir

---

## [Author Response · Author response to Decision Letter 1]

24 Dec 2024

Dear reviewer,

Thank you for your valuable feedback and comments on our manuscript titled “Genetic diversity and population structure of soybean (Glycine max (L.) Merril) germplasm”. We have carefully addressed all the points raised and made the necessary revisions as requested. We believed that these revisions have significantly improved the quality and clarity of the manuscript.

Thank you for your time and consideration

Best regards

Tenena SILUE

Ph.D. student

---

## [Decision Letter · Decision Letter 1]

2 Jan 2025

PONE-D-24-43093R1Genetic diversity and population structure of soybean (Glycine max (L.) Merril) germplasm.PLOS ONE

Dear Dr. SILUE,

Thank you for submitting your manuscript to PLOS ONE. After careful consideration, we feel that it has merit but does not fully meet PLOS ONE’s publication criteria as it currently stands. Therefore, we invite you to submit a revised version of the manuscript that addresses the points raised during the review process.

We look forward to receiving your revised manuscript.

Kind regards,

Tzen-Yuh Chiang

Academic Editor

PLOS ONE

Reviewers' comments:

Reviewer's Responses to Questions

**Comments to the Author**

1. If the authors have adequately addressed your comments raised in a previous round of review and you feel that this manuscript is now acceptable for publication, you may indicate that here to bypass the “Comments to the Author” section, enter your conflict of interest statement in the “Confidential to Editor” section, and submit your "Accept" recommendation.

Reviewer #1: (No Response)

2. Is the manuscript technically sound, and do the data support the conclusions?

Reviewer #1: No

3. Has the statistical analysis been performed appropriately and rigorously? 

Reviewer #1: (No Response)

4. Have the authors made all data underlying the findings in their manuscript fully available?

Reviewer #1: No

5. Is the manuscript presented in an intelligible fashion and written in standard English?

Reviewer #1: Yes

6. Review Comments to the Author

Reviewer #1: Dears …

The researchers analyzed the genetic diversity in soybean accessions using DArt technology. There isn't much new information available on the subject. The following adjustments are needed:

Abstract

The aim and the methods should be joined into a single phrase

It would be helpful to include a brief conclusion about the obtained results

Introduction

This section is too long and should be summarized

It is suggested that you use the most current references that you can find.

The authors should describe the DNA molecular marker related to this study

Detailed information regarding methodologies for genetic diversity should be provided.

The authors should provide the gap and hypothesis of this study

The authors should expressly indicate in a single sentence the originality of the work

What distinguishes this study is that the authors identified new tasks or linkages related to their investigation.

Both general and specific goals should be documented

Materials and Methods

The tissue type used for DNA extraction should be described

The name of commercial kit should be included

Each term must have its entire name provided.

It is essential to cite your sources or references for each approach

The manufacture of all instruments and materials should be stated

Why did not the authors perform all measurements of genetic diversity indices like Na, Ne, and I?

The authors should perform the computation of genetic diversity index for population

Results and discussion

This section is simply written and should be detailed

The results of tables should be detailed

All abbreviation should be written in full name

When describing the maximum and minimum values of the attributes, the authors should use scored data.

Every caption should be enhanced.

The Delta K figure of genetic structure should be spotted

The discussion did not have enough conviction. The analysis is a significant part of the remarks; yet, the majority of them merely repeat the findings without critically analyzing the facts. Please explain to readers how the authors' findings relate to those of other studies. Detailed information must be supplied on how each recorded parameter effects high or poor results. Their description should include all of the ramifications of setting high or low values for each of these parameters. The authors should interpret all measurements like PIC, He, and multivariate analysis

Conclusion

This section's authors should present a concise summary of the most important findings in an easy-to-read format. This area should be explored more in future studies.

Best regards

7. PLOS authors have the option to publish the peer review history of their article (what does this mean? ). If published, this will include your full peer review and any attached files.

**Do you want your identity to be public for this peer review?** For information about this choice, including consent withdrawal, please see our Privacy Policy .

Reviewer #1: **Yes: ** Nawroz Abdulrazzak Tahir

---

## [Author Response · Author response to Decision Letter 2]

21 Jan 2025

Authors’ Responses to Reviewers' comments

Reviewer #1: Dear …

The researchers analyzed the genetic diversity in soybean accessions using DArt technology. There isn't much new information available on the subject. The following adjustments are needed:

Abstract

1. The aim and the methods should be joined into a single phrase

Response: Thanks for your valuable observation. The aim and the methods have been joined into a single phrase. (Reference: Lines 22-25).

2. It would be helpful to include a brief conclusion about the obtained results

Response: This has been adequately taken care off (Lines 37-39).

Introduction

1. This section is too long and should be summarized

Response: Thank you for your observation. This section has been reduced to provide the most relevant information.

2. It is suggested that you use the most current references that you can find.

Response: Thank you for this comment, we have included as many recent references that are relevant to our work as possible while retaining the old references that are highly important in the revised version of the manuscript.

3. The authors should describe the DNA molecular marker related to this study

Response: Thank you for your observation, the requested information has been provided. (Lines 117-127) in the revised manuscript.

4. Detailed information regarding methodologies for genetic diversity should be provided.

Response: Thank you for your observation. The methodologies for assessing genetic diversity in crops was documented in line 68 – 83.

5. The authors should provide the gap and hypothesis of this study.

Response: Thank you for your comment. The requested information has been updated. (Lines 136-142).

6. The authors should expressly indicate in a single sentence the originality of the work

Response: Thank you for the comments. This aspect has been addressed. (Lines 142-144).

7. What distinguishes this study is that the authors identified new tasks or linkages related to their investigation.

Response: We appreciate your comments. We have included the statement showing how our work was unique following the identified gap in the IITA soybean breeding program.

8. Both general and specific goals should be documented

Response: Thank you for your comment. This observation has been provided (Lines 168-169).

Material and methods

9. The tissue type used for DNA extraction should be described

Response: Thank you for your comment. The description of the tissue type used for DNA extraction was documented in lines 182-184.

10. The name of the commercial kit should be included

Response: Thank you for your comment. We have included the names of the commercial kits as requested in the revised manuscript.

11. It is essential to cite your sources or references for each approach

Response: Thank you for your observation. Appropriate citations following the format of the PLOSONE journal have been made for the sources and approaches we explored and documented in the revised version of the manuscript.

12. The manufacture of all instruments and materials should be stated

Response: Thank you for your observation. We have tried our best to include the manufacturer of the instruments and materials that we could find as requested in the revised version of the manuscript. Our limitation was because the genotyping procedure (DNA extraction and genomic scan) were all performed by the DArT laboratory.

13. Why did not the authors perform all measurements of genetic diversity indices like Na, Ne, and I?

Response: we appreciate your contribution in ensuring our manuscript is better improved. Our understanding of Na, Ne, and I points to population size, effective sample size, and inbreeding depression, respectively which we do not believe may provide newer and different information than the diversity parameters (allele frequency, PIC, HO, He, 1/D, H’, and A) we have explored in the study.

14. The authors should perform the computation of genetic diversity index for population

Response: Thank you so much for this suggestion. We have included more population diversity assessment indices including the Shannon-Wiever index, inverse Simpson index, and Alpha diversity index to verify the results obtained from the marker diversity indices.

Results and discussion

15. This section is simply written and should be detailed

Response: Thanks so much for your observation. We have included more interpretations and tailored our discussion to the interpreted results in the revised version of the manuscript.

16. The results of tables should be detailed

Response: Thanks so much for your suggestion. The results in the tables have been improved to include a better caption and footnote adjustment in the revised version of the manuscript as requested.

17. All abbreviation should be written in full name

Response: Thank you for your comment. Abbreviations in the manuscript have been fully written as requested in the revised manuscript.

18. When describing the maximum and minimum values of the attributes, the authors should use scored data.

Response: Thank you for your comment. We have included minimum, mean, and maximum values in Table 1. The information in this table was drawn from quantitative data.

19. Every caption should be enhanced.

Response: Thank you for your observation. We have enhanced the captions in the Tables and figures within the revised manuscript as requested.

20. The Delta K figure of genetic structure should be spotted

Response

We appreciate your comment. We have incorporated the information of obtained in the manuscript and added the optimal cluster information from the silhouette plot for our HC as a supplementary file.

21. The discussion did not have enough conviction. The analysis is a significant part of the remarks; yet, the majority of them merely repeat the findings without critically analyzing the facts. Please explain to readers how the authors' findings relate to those of other studies. Detailed information must be supplied on how each recorded parameter effects high or poor results. Their description should include all of the ramifications of setting high or low values for each of these parameters. The authors should interpret all measurements like PIC, He, and multivariate analysis.

Response: Thank you so much for your observation. We have revised the discussion section of the manuscript to ensure it provides adequate conviction to the readers as requested in the revised manuscript.

Conclusion

22. This section's authors should present a concise summary of the most important findings in an easy-to-read format. This area should be explored more in future studies

Response: Thank you for your observation. We have provided the summary of the most important findings in a concise manner as requested.

---

## [Decision Letter · Decision Letter 2]

5 Mar 2025

Genetic diversity and population structure of soybean (Glycine max (L.) Merril) germplasm.

PONE-D-24-43093R2

Dear Dr. SILUE,

We’re pleased to inform you that your manuscript has been judged scientifically suitable for publication and will be formally accepted for publication once it meets all outstanding technical requirements.

Kind regards,

Tzen-Yuh Chiang

Academic Editor

PLOS ONE

Additional Editor Comments (optional):

Reviewers' comments:

Reviewer's Responses to Questions

**Comments to the Author**

1. If the authors have adequately addressed your comments raised in a previous round of review and you feel that this manuscript is now acceptable for publication, you may indicate that here to bypass the “Comments to the Author” section, enter your conflict of interest statement in the “Confidential to Editor” section, and submit your "Accept" recommendation.

Reviewer #2: All comments have been addressed

2. Is the manuscript technically sound, and do the data support the conclusions?

Reviewer #2: Yes

3. Has the statistical analysis been performed appropriately and rigorously? 

Reviewer #2: Yes

4. Have the authors made all data underlying the findings in their manuscript fully available?

Reviewer #2: Yes

5. Is the manuscript presented in an intelligible fashion and written in standard English?

Reviewer #2: Yes

6. Review Comments to the Author

Reviewer #2: Authors of the study Genetic Diversity and Population Structure of Soybean germplasm has exhausted the manuscript well.

However they must take note of the following;

1. L42-43 needs reference.

2. L 64-66 can be removed since the study is not about identifying genes.

3. L 117-118 seems the wording is not making the figures to add up well. Kindly rephase.

7. PLOS authors have the option to publish the peer review history of their article (what does this mean? ). If published, this will include your full peer review and any attached files.

**Do you want your identity to be public for this peer review?** For information about this choice, including consent withdrawal, please see our Privacy Policy .

Reviewer #2: **Yes: ** Emmanuel Amponsah Adjei

---

## [Editor Report · Acceptance letter]

PONE-D-24-43093R2

PLOS ONE

Dear Dr. SILUE,

I'm pleased to inform you that your manuscript has been deemed suitable for publication in PLOS ONE. Congratulations! Your manuscript is now being handed over to our production team.

Kind regards,

on behalf of

Dr. Tzen-Yuh Chiang

Academic Editor

PLOS ONE